# Comparing Zwitterionic and PEG Exteriors of Polyelectrolyte Complex Micelles

**DOI:** 10.3390/molecules25112553

**Published:** 2020-05-30

**Authors:** Jeffrey M. Ting, Alexander E. Marras, Joseph D. Mitchell, Trinity R. Campagna, Matthew V. Tirrell

**Affiliations:** 1Pritzker School of Molecular Engineering, University of Chicago, Chicago, IL 60637, USA; jting1@uchicago.edu (J.M.T.); marras@uchicago.edu (A.E.M.); jdm41297@gmail.com (J.D.M.); trinityc@uchicago.edu (T.R.C.); 2Center for Molecular Engineering and Materials Science Division, Argonne National Laboratory, Lemont, IL 60439, USA

**Keywords:** polyelectrolyte complexes, micelles, zwitterions, PEG, polymer chemistry

## Abstract

A series of model polyelectrolyte complex micelles (PCMs) was prepared to investigate the consequences of neutral and zwitterionic chemistries and distinct charged cores on the size and stability of nanocarriers. Using aqueous reversible addition-fragmentation chain transfer (RAFT) polymerization, we synthesized a well-defined diblock polyelectrolyte system, poly(2-methacryloyloxyethyl phosphorylcholine methacrylate)-*block*-poly((vinylbenzyl) trimethylammonium) (PMPC-PVBTMA), at various neutral and charged block lengths to compare directly against PCM structure–property relationships centered on poly(ethylene glycol)-*block*-poly((vinylbenzyl) trimethylammonium) (PEG-PVBTMA) and poly(ethylene glycol)-*block*-poly(l-lysine) (PEG-PLK). After complexation with a common polyanion, poly(sodium acrylate), the resulting PCMs were characterized by dynamic light scattering (DLS) and small angle X-ray scattering (SAXS). We observed uniform assemblies of spherical micelles with a diameter ~1.5–2× larger when PMPC-PVBTMA was used compared to PEG-PLK and PEG-PVBTMA via SAXS and DLS. In addition, PEG-PLK PCMs proved most resistant to dissolution by both monovalent and divalent salt, followed by PEG-PVBTMA then PMPC-PVBTMA. All micelle systems were serum stable in 100% fetal bovine serum over the course of 8 h by time-resolved DLS, demonstrating minimal interactions with serum proteins and potential as in vivo drug delivery vehicles. This thorough study of the synthesis, assembly, and characterization of zwitterionic polymers in PCMs advances the design space for charge-driven micelle assemblies.

## 1. Introduction

Polyelectrolyte complex assemblies have the unique ability to interface with an array of biologics [1,2,3,4]. This liquid–liquid phase separation process upon mixing oppositely charged polyelectrolyte solutions can be advantageous for partitioning biomolecules into compartmentalized domains [5,6,7,8]. When the complex domain is confined to the core of polyelectrolyte complex micelles (PCMs) with neutral-charged block polycation and/or polyanion architectures, therapeutically relevant cargo can be captured, protected by the outer hydrophilic corona and released in response to changes in pH or ionic strength. In terms of polymer selection for the outer stealth layer, poly(ethylene glycol) (PEG; synonymously polyethylene oxide or PEO) is one of the most common ways to facilitate steric stabilization from aggregation and facilitate biocompatibility. PEG-containing PCMs can self-assemble into discrete nanoparticles with controlled size and morphologies at dilute polymer conditions [9,10] and resemble ordered gels at higher polymer concentrations [11,12,13,14]. However, certain limitations of PEG have become identified in the past decade. For biologic applications, this includes hypersensitivity after drug administration, deteriorating oxidative stability over time, and accelerated systemic clearance upon repeated dosage [15,16,17]. Thus, there has been substantial effort to explore PEG alternatives that can instill colloidal stability in PCM carriers in aqueous environments while mitigating some of these drawbacks [18,19,20,21,22,23].

One promising alternative includes polyampholytes, or polyelectrolytes with both cations and anions along the chain. Polyzwitterions are polyampholytes with paired cation and anion moieties per repeat unit along the electrostatically neutral chain [24,25]. The 2-methacryloyloxyethyl phosphorylcholine (MPC) methacrylate monomer, in particular, has become transformative as a building block for polyzwitterions in emerging biomaterials and biodevices [26], as the phospholipid polar groups provide hydration, but does not disturb the hydrogen bonding of water [27]. When polymerized to poly(2-methacryloyloxyethyl phosphorylcholine methacrylate) (PMPC), the zwitterionic phosphorylcholine moieties endow high resistance to nonspecific protein adsorption, cellular adhesion and blood coagulation, as well as cell membrane penetration, molecular recognition, and underwater lubrication [28,29]. These bioinert “stealth” properties were exploited in antifouling [30] and water treatment [31] materials, bioengineering [32], and nanomedicine applications [33]. Furthermore, the anti-polyelectrolyte effect, where certain polyzwitterion chains swell (increased coil size and viscosity) with added salt [34,35,36], was shown to not affect the solution conformation of PMPC [37,38]. These combined attributes have allowed PMPC to emerge as a versatile material platform for a diverse array of bioapplications.

When adapted to hydrophilic coronas in nanocarriers, several reports that have directly compared PMPC to PEG have shown enhanced delivery for siRNA polyplexes [33], bovine serum albumin protein nanoparticles [39], α-chymotrypsin proteins [40], and papain enzyme bioconjugates [41]. These PMPC-containing block polymers are often synthesized using reversible-deactivation radical polymerization (RDRP). Lowe and McCormick provide an excellent review on the controlled synthesis of zwitterionic polymers [42], followed by more recent examples of PMPC systems in the literature prepared using atom transfer radical polymerization (ATRP) [43,44,45,46] or reversible addition-fragmentation chain transfer (RAFT) polymerization [33,47,48]. Thus, the use of RDRP to produce PMPC block polyelectrolytes for fundamental self-assembly investigations can lead to more efficacious PCM vehicles in terms of biomolecule storage and delivery. For example, Takahashi and coworkers have reported on how NaCl can trigger morphologic transitions in PCM assemblies comprising RAFT PMPC block polyelectrolytes [49]. These results have advanced our overall understanding of how to tailor the static and dynamic properties of polyelectrolyte complex assemblies.

The objective of this work is to evaluate and compare the formulation and stability of polyelectrolyte-based nanocarriers comprising PMPC and PEG as hydrophilic coronas and chemically distinct charged cores. Using aqueous RAFT polymerization, we have previously established a synthetic framework to prepare designer block polyelectrolytes from PEG macromolecular RAFT chain transfer agents [50]. Specifically, the diblock polycation PEG-*block*-poly((vinylbenzyl)trimethylammonium chloride) (PEG-PVBTMA) was paired with various synthetic polyanion architectures, including poly(sodium acrylate) (PAA) [50,51,52,53], PEG-*block*-poly(styrene sulfonate) [50,54], and DNA oligonucleotides [55] to form assemblies of varying size, shape, and responsivity to salt. Because of the ease and versatility of RAFT chemistry for the production of stimuli-responsive materials [56], these endeavors have enabled us to establish structure–property relationships with special attention to effects of modulating charged block length, chemical identity, and polyelectrolyte pairing. Figure 1 depicts the extension of this molecular engineering effort by comparing PCMs of PAA with three diblock polycations: PEG-*block*-poly(l-lysine) (PEG-PLK), PEG-PVBTMA, and PMPC-PVBTMA. For ease of nomenclature and distinction we refer to PMPC-containing systems as zwitterionic PCMs (zPCMs).

## 2. Materials and Methods

### 2.1. Materials

Certain commercial equipment and materials are identified in this study in order to specify adequately the experimental procedure. In no case does such identification imply recommendations by the National Institute of Standards and Technology (NIST) nor does it imply that the material or equipment identified is necessarily the best available for this purpose.

The following chemicals were reagent grade and used as received unless otherwise specified: poly(ethylene glycol)-*block*-poly(lysine) (Alamanda Polymers; Huntsville, AL, USA), poly(ethylene glycol) methyl ether (2-methyl-2-propionic acid dodecyl trithiocarbonate) (Sigma, St. Louis, MO, USA, Reported *M*_n_ 10,000 and 5000 g/mol; verified experimentally) with *Đ* = 1.02 and 1.01, respectively), 2-methacryloyloxyethyl phosphorylcholine (MPC, Sigma, St. Louis, MO, USA, 97%), 4-cyano-4-(phenylcarbonothioylthio)pentanoic acid N-succinimidyl ester (CPSE, Sigma, St. Louis, MO, USA), 2-(butylthiocarbonothioylthio)propanoic acid (BuPA, Boron Molecular, Raleigh, NC, USA, ≥95%), sodium acrylate (NaA, Sigma, St. Louis, MO, USA, 97%,), (vinylbenzyl)trimethylammonium chloride (VBTMA, Sigma, St. Louis, MO, USA, 99%), 2,2’-azobis [2-(2-imidazolin-2-yl)propane] dihydrochloride (VA-044, Wako Chemicals, Richmond, VA, USA), acetic acid (glacial, Sigma, St. Louis, MO, USA, ≥99.85%), sodium acetate trihydrate (Sigma, St. Louis, MO, USA, ≥99%), sodium chloride (Sigma, St. Louis, MO, USA, ≥99.0%), trifluoroacetic acid (TFA, ReagentPlus^®^, Sigma, St. Louis, MO, USA, 99%), sodium nitrate (NaNO_3_, ACS reagent, Sigma, St. Louis, MO, USA, ≥99.0%), sodium azide (NaN_3_, ReagentPlus^®^, Sigma, St. Louis, MO, USA, ≥99.5%), Fetal Bovine Serum (ES Cell-Qualified, Thermo Fisher, Waltham, MA, USA), and SnakeSkin dialysis tubing (MWCO 3.5K, 22 mm, Thermo Scientific, Waltham, MA, USA).

For RAFT polymerizations, monomers VBTMA and NaA were used as received. MPC monomer was filtered with diethyl ether to remove trace inhibitor, vacuum dried and stored under dried nitrogen until use. Acetate buffer solution (pH 5.2) was prepared from 0.1 M acetic acid and 0.1 M sodium acetate trihydrate (42/158, *v*/*v*). Water was filtered from a Milli-Q water purification system at a resistivity of 18.2 MΩ cm at 25 °C.

### 2.2. RAFT Synthesis of PEG-PVBTMA, PAA and PMPC-PVBTMA

We synthesized PEG-PVBTMA with aqueous RAFT polymerization from a PEG macromolecular chain transfer agent (macro CTA), according to parallel synthesis and characterization procedures described by Ting et al. [50] previously. PAA was synthesized using BuPA as a RAFT chain transfer agent (CTA) in aqueous settings and characterized. Detailed molecular characterization is provided in the Appendix A.

Scheme 1 depicts the synthesis of PMPC and PMPC-PVBTMA. The following describes a representative RAFT protocol: To a dried round bottom flask, the chemical precursors (monomer, RAFT CTA and VA-044 initiator) were combined in solvent and sealed. A ratio of 10:1 equivalence of RAFT CTA to initiator ([CTA]_0_/[I]_0_ = 10:1) was used. The flask was degassed for at least 30 min with dried nitrogen sparging, before running the reaction at 50 °C under constant stirring for >18 h. The final monomer conversion was confirmed by ^1^H NMR spectroscopy before quenching by cooling to room temperature and opening the reactor to air. The crude red–orange mixture was dialyzed against Milli-Q water for 3 cycles of 8 h each, lyophilized, and characterized by size-exclusion chromatography with multiangle light scattering, ^1^H NMR spectroscopy, and thermogravimetric analysis.

### 2.3. Size-Exclusion Chromatography with Multiangle Light Scattering (SEC-MALS)

SEC measurements were conducted on a Waters liquid chromatogram (Waters Corporation, Milford, MA, USA; consisting of well-stirred mobile phase reservoir, Waters In-Line Degasser AF, Waters 1515 Isocratic high-performance liquid chromatography (HPLC) pump, Waters 2707 Autosampler) equipped with three Waters columns (Ultrahydrogel Column 500, Ultrahydrogel Column 120, and Ultrahydrogel Column 250) maintained at 35 °C with pore sizes suitable for materials with effective molecular weights from 1000 to 400,000 g/mol. The SEC is equipped with a Waters 2998 photodiode array detector (Waters Corporation, Milford, MA, USA), a miniDAWN TREOS II multi-angle laser light scattering (MALS) detector (Wyatt Technology Corporation, Santa Barbara, CA, USA) at a laser wavelength of 658 nm (3 angles), and a Wyatt Optilab T-rEX refractive index detector (Wyatt Technology Corporation, Santa Barbara, CA, USA). Empower 3 and ASTRA^®^ 6 Software were used for setting up the experiments and analyzing data, respectively.

Samples were dissolved and immediately run. For PEG and PMPC, 0.1 M NaNO_3_ + 0.01% NaN_3_ was run as the mobile phase at 1.0 mL/min at 35 °C. For PAA, 0.3 M NaNO_3_ + 0.01 M NaH_2_PO_4_ at pH 7 was run as the mobile phase at 1.0 mL/min at 25 °C. For diblock cationic polyelectrolytes, a mixture of acetonitrile and water (40/60% *v*/*v*) with 0.1% trifluoroacetic acid was run as the mobile phase at 1.0 mL/min at 35 °C. The change in refractive index value (*dn*/*dc*) values were determined using an Abbe refractometer with a red light-emitting diode as a light source in the appropriate SEC mobile phase at 25 °C, in which refractive index values at various polymer concentrations were collected in triplicate (Appendix A). The *dn*/*dc* of each block polyelectrolyte (*dn*/*dc*)_*block*_ was calculated using Equation (1), which assumes that the (*dn*/*dc*)_*block*_ is a weighted linear combination (i.e., the weight fraction of each block) of the change in each refractive index value of individual polymers *A* and *B*.
(1)$${\left(\frac{dn}{dc}\right)}_{block}=w_{A}{\left(\frac{dn}{dc}\right)}_{A}+\left(1-w_{A}\right){\left(\frac{dn}{dc}\right)}_{B}$$

### 2.4. Proton Nuclear Magnetic Resonance (^1^H NMR) Spectroscopy

^1^H NMR measurements were collected using a Bruker AVANCE III HD 400 MHz NanoBay spectrometer (Bruker Corporation, Billerica, MA, USA) with 16 transients to minimize signal-to-noise. ^1^H NMR spectra were processed and analyzed using iNMR (Version 5.5.7).

A representative ^1^H NMR spectrum of the PMPC homopolymer is shown in Figure 2. ^1^H NMR (400 MHz, D_2_O) δ: 0.6–1.3 ppm (2H,–C**H_2_**–), 1.6–2.2 ppm (3H,–C**H_3_**), 2.9–3.2 ppm (9H,–N-(C**H_3_**)_3_), 3.3–3.6 (2H,–C**H_2_**-N–) and 4.0–4.3 (6H,–C**H_2_**-O–).

A representative ^1^H NMR spectrum of the PMPC-*b*-PVBTMA homopolymer is shown in Figure 3. The calculated chemical composition of VBTMA was determined to be 75%, in good agreement with the targeted chemical composition of VBTMA (68%). ^1^H NMR (400 MHz, D_2_O) δ: 0.7–2.1 ppm (alkyl backbone), 2.6–3.0 ppm (9H,–N-(C**H_3_**)_3_), 3.0–3.3 ppm (9H,–N-(C**H_3_**)_3_), 3.5–3.7 ppm (2H,–C**H_2_**-N–), 3.8–4.5 ppm (6 H,–C**H_2_**-O–; 2H,–C**H_2_**-N–), and 6.1–7.4 ppm (4H, Ar**H**).

### 2.5. Thermogravimetric Analysis (TGA)

TGA was conducted using a TA Instruments (New Castle, DE, USA) Discovery TGA equipped with an infrared furnace, auto-sampler and a gas delivery module. Nitrogen was used as the purge gas set at a flow rate of 10 mL/min; a heating rate of 10 °C/min was set for all samples. TA TRIOS software (Version 2.2) was used to analyze the thermal transitions.

### 2.6. Micelle Preparation

Micelles were prepared via rapid mixing of water, NaCl or MgCl_2_, and polyelectrolyte solutions. Water and salt were first mixed with the neutral-cationic diblock polymer and vortexed, then the polyanion was added followed by additional rapid mixing. The polyelectrolytes were added from a 5 mg/mL stock for a final concentration of 3 mM charge concentration for both the cation and anion. For PEG-PLK and PEG-PVBTMA systems, reversing the order had no effect on the final assembly structure. For PMPC-PVBTMA, we unexpectedly observed a dependency of adding the diblock polycation to the polyanion (or vice versa) on the micelle size and response to salt, as shown in Appendix A. The origin of this phenomenon is the subject of ongoing investigation.

### 2.7. Dynamic Light Scattering (DLS)

Dynamic light scattering (DLS) measurements were made using a Brookhaven Instruments (Holtsville, NY, USA) BI-200SM Research Goniometer System with an incident laser (λ = 637 nm) at room temperature. A dust-free decalin bath was used to match the refractive index of glass. The angular dependence of *D* was acquired by plotting *q*^2^ versus decay rate Γ. A linearity over a range of scattering angles is a good indication of isotropic scatters. The average hydrodynamic radius of scatterers under Brownian diffusion was calculated by the Stokes–Einstein relationship. The correlation function at each angle was fitted to a first-order single exponential in MATLAB (Version R2019a). The size distribution was obtained using the REPES (regularized positive exponential sum) algorithm [57]. Kinetics experiments were done on the same instrument by recording data every 5 min for up to 16 h in a sealed sample container to minimize evaporation.

The low salt (0–200 mM NaCl) screening studies (Appendix A) were carried out on a 163.5° single-angle Wyatt Möbius instrument (Santa Barbara, CA, USA) with an incident laser (λ = 532 nm) at 20 °C. Prepared micelle solutions were directly transferred to quartz micro-cuvettes. The size and colloidal polydispersity index were analyzed using DYNAMICS software (Version 7).

### 2.8. Small-Angle X-ray Scattering (SAXS)

Small-angle X-ray scattering (SAXS) measurements were made at beamline 12-ID-B of the Advanced Photon Source (APS) at Argonne National Laboratory (Lemont, IL, USA) and at beamline 4–2 of the Stanford Synchrotron Radiation Lightsource (SSRL) at SLAC National Accelerator Laboratory (Menlo Park, CA, USA). Glycerol was added to all samples for a final concentration of 1% by volume to prevent sample degradation.

For APS experiments, micelle samples were irradiated in a thin-wall glass capillary flow cell with a photon energy of 14 keV. Data were reduced in MATLAB at the beamline. Background subtraction and fitting were performed using the multi-level modeling macros distributed with the Irena software package (Release 2.67) [58] for Igor Pro (Version 7.08) as described in Ref [59].

For SSRL experiments, the sample-to-detector distance was set to 3.5 m with measurements collected at a photon energy of 9 keV. Aliquots of 30 µL of the polyelectrolyte micellar suspension were loaded onto the automated fluid sample loader at the beamline. At least 20 consecutive 1 s exposures were collected first from the buffer background (water or water/salt mixtures), followed by samples. During data collection, solutions were oscillated in a stationary quartz capillary cell to maximize the exposed volume and reduce the radiation dose per unit volume. The SAXS data were radially integrated, analyzed for radiation damage, and buffer subtracted using the automated data reduction pipeline available at the beam line. Only data that did not show any evidence of radiation damage were included in the final average for each sample.

### 2.9. Micelle Salt Dependence

Micelles were prepared in numerous salt solutions and compared via SAXS and DLS. High salt concentrations disrupt complex formation as large amounts of counterions compete with ion pairing. This is noticeable as SAXS and DLS intensity decrease and nanoparticles lose shape and dissolve into solution. The salt resistance of a polyelectrolyte complex can be used as a measure of ion pair stability in the assembly material. Micelles were assembled in reported ionic conditions.

### 2.10. Micelle Stability Tests

Micelle solutions were prepared in water with no salt at 4.5 mg/mL of total polymer. These solutions were then diluted 5× in 100% fetal bovine serum (FBS) and DLS measurements were taken at *θ* = 90° scattering angle at room temperature for 2 min collection time every 5 min. The autocorrelation function and light scattering intensity were recorded for at least an 8 h period to observe changes in the size of detectable particles in the presence of FBS.

## 3. Results and Discussion

### 3.1. PMPC-PVBTMA Synthesis and Characterization

We have previously demonstrated the consequences of modifying the chemical nature of the core in PCMs for the complexation of oligonucleotides with PLK or PVBTMA units [55]. Other studies that employ block polyelectrolytes with PEG as the neutral block have also highlighted analogous design considerations [13,60]. However, far fewer PCM studies have focused on the fundamental PCM properties of zwitterionic block polyelectrolytes that form core-shell structured micelles. To this end, we sought to prepare direct polycation analogs to PEG-PLK and PEG-PVBTMA and evaluate aspects of PCM formulation with PAA as a model polyanion. For nomenclature throughout this work, subscripts next to the neutral and charged polymer block represent the molar mass and number-average degree of polymerization, respectively.

As shown in Scheme 1, the RAFT synthesis of PMPC-PVBTMA diblock polyelectrolytes were achieved using CPSE as a suitable RAFT CTA for MPC monomer in deoxygenated methanol at 50 °C, followed by chain extension of the PMPC macromolecular CTA with VBTMA monomer in deoxygenated acetate buffer/ethanol at 50 °C. Each reaction was run overnight (>18 h). The experimentally measured degree of polymerization (DP) of the charged block was similar to the targeted values, affording PMPC_5K_-PVBTMA_50_ and PMPC_10K_-PVBTMA_97_. ^1^H NMR studies indicated that the RAFT synthesis in both steps was successful. In addition, aqueous SEC-MALS showed distinct monomodal peaks of the PMPC precursor and PMPC-PVBTMA diblocks in the optimized mobile phase solutions. Representative SEC refractive index traces are shown in Figure 4. We observe a very small fraction of unreacted PMPC by SEC-MALS, which likely represents dead chains that lost end-group fidelity.

Table 1 summarizes the characterization of the polymer number—average molecular weight (*M*_n_), experimentally measured degree of polymerization (DP) of the charged blocks by both ^1^H NMR spectroscopy and SEC-MALS, dispersity (*Đ*), and degradation temperature (*T*_d_). Data on the PEG-PLK samples are reported as received. For all diblock polyelectrolyte systems, the *M*_n_ distribution remained relatively narrow (*Đ* < 1.2). Relatively good agreement between DP values was observed. By thermogravimetric analysis, two distinct thermal degradations corresponded to each neutral and charged block (Appendix A). Thus, the RAFT chain extension protocols for PEG-PVBTMA and PMPC-PVBTMA were controlled and successful in generating neutral-charged diblock polyelectrolytes. The PAA polyanion was also synthesized with aqueous RAFT chemistry targeting a DP of 50 and fully characterized (*M*_n_ = 4.9 and 5.0 kg/mol by SEC-MALS and NMR end-group analysis, respectively; *Đ* = 1.11). Details of the RAFT synthesis and characterization of PAA are described in the Supplementary Information. In short, these six uniform diblock polycations were used for complexation with PAA to prepare PCMs for solution studies.

### 3.2. Polyelectrolyte Complex Micelle (PCM) and Zwitterionic PCM (zPCM) Assembly, Salt Resistance, and Stability

In order to elucidate the effect these physical and chemical differences have on micelle structure, we used the six polymers from Table 1 to separately complex with PAA_50_ as a common polyanion, to form micelles in water. PCMs and zPCMs were assembled via direct dissolution and rapid mixing into the desired salt solution. Three polymer chemistries are being compared (PEG-PLK, PEG-PVBTMA, and PMPC-PVBTMA) at two lengths each (i.e., a 10 kg/mol neutral block with ~100-unit cation versus a 5 kg/mol neutral block and ~50-unit cation). For nomenclature, these systems will be referred to as neutral_10k_-charged_100_ and neutral_5k_-charged_50_. First, we screened the assembly hydrodynamic radius (*R*_h_) and size distribution of PEG-PVBTMA and PMPC-PVBTMA systems as a function of salt with a single-angle DLS instrument. In a low salt regime from 0–75 mM NaCl, no structural differences were detected, indicating strongly associated complexes (Appendix A). At 100 mM NaCl, we observed an increase in the polydispersity of the zPCMs only, but minimal change in radius for both neutral_10k_-charged_100_ and neutral_5k_-charged_50_ lengths. In comparison, the PCMs for both systems with an outer PEG corona exhibited no change in aggregation through 200 mM NaCl. These results suggest that increased salt conditions may induce aggregation or weaken complex assemblies to the point that they lose their well-defined exterior when PMPC corona is used, but not for a PEG corona.

To better understand this salt-driven transition in PCM and zPCM assembly, multi-angle DLS and SAXS were used to quantify the micelle shape and the size of the core and corona more precisely, providing complete structural morphology for each system. Table 2 shows the salt-free DLS characterization of the size and distribution for the PCMs and zPCMs between 60–120° in 15° increments. The *R*_h_ and colloidal polydispersity (PDI = μ_2_/ Γ^2^) were determined by fitting the autocorrelation function to a cumulant expansion [61] and via regularized positive exponential sum (REPES) analysis [57]. See the Appendix A for the angular dependence of the fitted autocorrelation functions and the regression of Γ versus q^2^ data. Excellent agreement in micelle size and low polydispersity between both methods were reported, again with a consistent hydrodynamic size difference where PMPC-PVBTMA was the largest and PEG-PLK was the smallest. The zPCMs exhibited a slightly larger hydrodynamic volume relative to the PCM counterparts, which is attributed to water solvating phosphorylcholine groups in the corona shell [26].

Figure 5 shows DLS and SAXS results for each neutral_10k_-charged_100_ system in 100 mM NaCl. For all six systems we observe micelle formation at 100 mM NaCl. The *R*_h_ for all systems was ~30 nm by REPES under this low salt condition. When complexed with PAA, PEG-PVBTMA and PMPC-PVBTMA systems form spherical micelles, apparent from the flat (scattering intensity ~ q^0^) slope at low q (q ≤ 0.01), while PEG-PLK appears to form worm-like micelles, exhibiting a characteristic power law ~q^-2^ dependence [55,62]. Fitting SAXS data provides a Guinier radius (*R*_Guinier_), which is smaller than *R*_h_, as the polymer-dense core dominates scattering. For PCMs with a PEG corona, *R*_Guinier_ is predominately the micelle core radius, as PEG has minimal scattering contrast over background. PMPC, however, has heavier phosphorus atoms that will scatter more than PEG and likely influence *R*_Guinier_ as zPCMs. We notice a consistent difference in *R*_Guinier_ where PMPC-PVBTMA is the largest and PEG-PLK is the smallest, displayed in Table 2, consistent with DLS screening results. The presence of PMPC likely contributes to a larger *R*_Guinier_ due to increased corona scattering and to a larger *R*_h_ (and larger, more hydrated corona) due to the disruption of hydrogen bonding formation between water molecules [26]. Furthermore, longer cationic blocks formed larger micelle cores throughout this study, consistent with previous reports [55,59,63] of micelle size being solely dependent on the charged block length of the block polymer and independent of homopolymer length.

Next, we aimed to evaluate the salt resistance and stability of the prepared PCMs and zPCMs. Recent reports have emphasized the importance of the charged group and neutral block chemistry for influencing complex micelle stability. In general, stability can be quantified by salt resistance—the critical salt concentration at which a complex assembly comes apart and dissolves into solution due to the abundance of ions in solution. Electrostatic interactions are screened, thereby reducing the effective charge density holding the nanoparticle assembly together. In the individual polyelectrolyte solutions of PEG-PLK, PEG-PVBTMA, and PMPC-PVBTMA, there was no observed difference in the disappearance of the signature correlation peak that is prominent in the SAXS profile by adding up to 500 mM NaCl (Appendix A). This suggests that the conformation of the neutral-charged polycation chains respond to salt in a similar manner. In order to probe the stability of these systems and their behavior in higher concentrations of monovalent and divalent salts, we formed PCMs and zPCMs in 100, 250, and 500 mM NaCl, as well as 50 and 100 mM MgCl_2_.

Figure 6 shows the SAXS results for the neutral_10k_-charged_100_ cases and fitting results are available in Table 3. The data for neutral_5k_-charged_50_ systems show similar trends and can be found in Appendix A. For PEG-PLK PCMs, minimal structural differences or changes in scattering intensity were seen across all conditions, signifying strong complexation. PEG-PVBTMA formed PCMs at 100 and 250 mM NaCl, as well as 50 mM MgCl_2_ but not at higher salts. PMPC-PVBTMA exhibited the weakest complexation, with well-formed zPCMs only at 100 mM NaCl. Additionally, both PVBTMA systems revealed a loss in intensity and increase in size dispersity with increased salt, which was not present in the PEG-PLK system. We have shown previously that PLK forms stronger ion pairs with DNA compared to PVBTMA [55], and here we observe a consistent trend with PAA. Overall, we see a drastic discrepancy in salt stability due to the neutral versus zwitterionic block. PMPC-PVBTMA is unable to form micelles at 50 mM MgCl_2_ and forms extremely polydisperse poorly formed micelles at 250 mM NaCl, conditions that are favorable for micelle formation when using the PEG variant. Additional SAXS data and models are available in Appendix A.

### 3.3. PCMs and zPCMs in Biologically Relevant Conditions

Since micelle stability is crucial for systemic circulation and controlled release, we investigated the structural integrity of prepared PCMs and zPCMs in FBS. The temporal evolution of the micelles in full serum was monitored with static scattering intensity (*I*). As previously shown by Lin and coworkers [64], this technique provides a straightforward way to detect micelle-protein interactions since *I* is proportional to *KcM_w_*, where *K* is the prefactor constant from the optics setup, *c* is the sample mass concentration, and *M*_w_ is the weight-average molecular weight of the suspended sample. The amount of light scattered is directly proportional to the molar mass and concentration of nanoparticles in solution. Because the pristine micelles that were prepared from all polyelectrolyte pairings are larger than the serum proteins in the FBS control sample (Appendix A), the total I will increase if the *M_w_* increases as a result of strong associative interactions between proteins and micelles.

For representative neutral_10k_-charged_100_ PEG-PLK, PEG-PVBTMA, and PMPC-PVBTMA micelles in FBS, Figure 7 provides a summary of the average change in *I* over the course of 8 h, along with the apparent size distribution at 1 and 10 h. Experiments were conducted multiple times to ensure repeatability (see Appendix A for repeats). The FBS negative control showed no evidence of aggregation or sedimentation over 8 h. It should be noted that FBS exhibits a broad bimodal distribution, so the PCMs cannot be distinguished from serum proteins. In the stability tests, the total *I* for both the PEG-PLK and PEG-PVBTMA systems remained relatively constant, while the PMPC-PVBTMA system increased by approximately 10% in *I* on average with greater variation over time. REPES analysis of the PMPC-PVBTMA system shows the growth of ~200 nm aggregates over 1 to 10 h. This slight stability issue may be mitigated by increasing the degree of polymerization of PMPC, as the total length of the zwitterionic corona in micelles were previously shown to affect nanoparticle stabilization and delivery efficacy [33]. Nevertheless, because protein adsorption onto nanoparticle carriers in biological settings is a known design consideration for micelle assemblies [65,66], the prepared PCMs and zPCM show good overall resistance to destabilization by serum.

## 4. Conclusions

In alignment with greater efforts reported recently on elucidating the interplay between electrostatics and other non-covalent associations on polyelectrolyte complexation [67,68], this work provides a central platform for designing charged micelle-based nanocarriers with fundamental structure–property relationships. Neutral blocks in PCMs and zPCMs force nanophase separation and provide a protective corona around a charged polymer core, which often contains a sequestered cargo. Understanding the physical effects attributed to zwitterionic and neutral polymer coronas and chemically distinct charged polymer cores, expands the design space and versatility of PCMs and zPCMs. Altogether, we have prepared a series of polyelectrolyte-based nanocarriers from PEG-PLK, PEG-PVBTMA, and PMPC-PVBTMA with PAA, characterized the self-assembled micelle structure in water as a function of salt, and evaluated long-term stability in biologically relevant conditions. Aqueous RAFT polymerization was used to synthesize a well-defined zwitterionic diblock polyelectrolyte system PMPC-PVBTMA at various block lengths. Upon complexation with PAA, at 100 mM NaCl the zPCMs exhibited spherical micelles by DLS and SAXS with *R*_h_ ~30 nm and *R*_Guinier_ ~20 nm. These uniform assemblies were slightly larger than the PCMs formed from PEG-PLK and PEG-PVBTMA, attributed to the highly solvated PMPC corona that accounts for the difference between the lengths determined by a hydrodynamic volume and Guinier approximation for each scattering technique, respectively. Scattering data are available through the Materials Data Facility [69,70] at doi:10.18126/n4un-6usf [71].

Characterizing micelle structure and disassembly in the presence of monovalent and divalent salts provide a measure of complex stability. Our results show that complexes formed by PEG-PLK are much more stable against salt, followed by PEG-PVBTMA which is slightly more resistant to salt than PMPC-PVBTMA, the least stable system. This result is consistent for both neutral_5k_-charged_50_ and neutral_10k_-charged_100_ block lengths. Overall, this study examines neutral and zwitterionic chemistries used as polymer coronas in PCMs to protect sequestered cargo and to control particle size, stability, and environmental responsiveness.

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
