# Peer review of "Comparing Zwitterionic and PEG Exteriors of Polyelectrolyte Complex Micelles"

_molecules, 2020, doi:10.3390/molecules25112553_

Round 1

Reviewer 1 Report

The manuscript entitled »Comparing Zwitterionic and PEG Exteriors of Polyelectrolyte Complex Micelles« written by J. M. ting et. al. presents a systematic study of stability of complex micelles. Effects of chemical structure and charge distribution of micelle building blocks were investigated.

The manuscript is well-written and provides sufficient insight of the conducted work, including description of compound synthesis and characterization. Results are interesting and are clearly presented. Several experimental techniques have been used throughout the study ranging from scattering techniques, NMR, and thermogravimetry.

After reading the manuscript I have only two suggestions to authors:

  • the “Discussion” section is rather short compared to previous sections, and the “Conclusions” part is missing
  • effects of salt concentration and type on micelle stability should be explained in more detail

I therefore recommend the publication of the manuscript in the »Molecules« journal after addressing minor issues listed above.

Author Response

We thank the reviewer for the critical review and feedback.

The final section was inadvertently mislabeled; we have changed section 3 from “Results” to “Results and Discussion” (since both data and interpretation are presented) and section 4 from “Discussion” to “Conclusions” (since this is an overall summary of the findings and impact of this work).

We have included additional description of the salt concentration and type with further discussion in Section 3.2 regarding micelle salt resistance

Reviewer 2 Report

The paper entitled "Comparing zwitterionic and PEG exteriors of polyelectrolyte complex micelles" by Ting et al. was well designed to investigate the characterize the charge-driven micelle assemblies. This study was performed using with well-defined polymer structure, and the results will provide suitable information.

Author Response

We thank the reviewer for the positive feedback

Reviewer 3 Report

In this interesting work a series of polyelectrolyte-based nanocarriers from PEG-PLK, PEG-PVBTMA, and PMPC-PVBTMA with PAA have been prepared and characterized by different methods. The self-assembled micelle structure in water was studied as a function of salt concentration, and evaluated long-term stability in biologically relevant conditions.

I would like to clarify some questions, see below:

  1. Line 282 Table 1. How much the measured molecular masses are consistent with the subscripts of the systems, for instance PMPC10K-PVBTMA97, shown in the table?

  1. Line 368 I ~ KcMw   (2) it is not equation, it is only proportionality.
  2. Table 2 “Aspect ratio” - What is means?

  1. It is interesting to discuss the differences in the sizes of the PMPC-PVBTMA copolymer (Fig. 5) and the sizes of PMPC-PVBTMA / PAA micelles (Table 2). If we go to the volume, then such a difference in the radii will lead to a 100-fold change in volume.
  1. From conclusion “Upon complexation with PAA, the zPCMs exhibited ~40 nm spherical micelles. These uniform assemblies were slightly larger than the PCMs formed from PEG-PLK and PEG-PVBTMA (~20-30 nm).“ However, in the text, not in the table, on the graphs, such large values for the studied micelles are not observed.

  1. Why are you sure that the formed micelles are only spherical in shape?

  1. SM S.4 “… dn/dc of the PMPC5K-PVBTMA50 and PMPC10K-PVBTMA100 to be 0.1661 and 0.1663 mL/g, respectively.”

I believe that with the same composition of the copolymers, the calculated values dn/dc should coincide. It is unlikely that calculated values of dn/nc are determined with an accuracy of 0.0001 mL/g. Three significant numbers are enough.

Author Response

Reviewer: 

In this interesting work a series of polyelectrolyte-based nanocarriers from PEG-PLK, PEG-PVBTMA, and PMPC-PVBTMA with PAA have been prepared and characterized by different methods. The self-assembled micelle structure in water was studied as a function of salt concentration, and evaluated long-term stability in biologically relevant conditions. I would like to clarify some questions, see below:

Response: We thank the reviewer for the helpful suggestions for improving this manuscript. A point-by-point response to these questions are provided below.

Line 282 Table 1. How much the measured molecular masses are consistent with the subscripts of the systems, for instance PMPC10K-PVBTMA97, shown in the table?

Response: The reviewer raises an important point. We have updated Table 1 with the calculated degree of polymerization (DP) of the charged block, experimentally measured by 1H NMR and SEC. This provides readers a way to assess the characterization from both techniques, which are in relatively good agreement. Because of the propagation of error associated with accurately measuring dn/dc, we believe that the DP by 1H NMR is more accurate and maintain its use as nomenclature throughout the manuscript.

We also note that upon reviewing this data, there were two typographical mistakes discovered for the PMPC5K-PVBTMA50 and PMPC10K-PVBTMA97 entries (inconsistent with the provided SEC traces). These values have been corrected.

Line 368 I ~ KcMw (2) it is not equation, it is only proportionality.

Response: The reviewer is correct. We have corrected this statement to “…since I is proportional to KcMw, where K is the prefactor constant from the optics setup, c is the sample mass concentration, and Mw is the weight-average molecular weight of the suspended sample,” which was described by Lin and coworkers in Reference 63.

Table 2 “Aspect ratio” - What is means?

Response: The aspect ratio is a parameter of the spheroid form factor. We have the following definition to the Table 2 as a footnote for the Aspect Ratio column. “Aspect ratio of an oblate spheroid is the ratio of the principal axis and the diameter perpendicular to this axis; an aspect ratio of 1 denotes a perfect sphere.”

It is interesting to discuss the differences in the sizes of the PMPC-PVBTMA copolymer (Fig. 5) and the sizes of PMPC-PVBTMA / PAA micelles (Table 2). If we go to the volume, then such a difference in the radii will lead to a 100-fold change in volume.

Response: These radii are actually referring to different dimensions. Dynamic light scattering provides the hydrodynamic radii (Rh), meaning the entire radius of the micelle (core+corona), while SAXS predominately scatters from the polymer-dense core, providing the radius of the core only (Rc). We have updated the variable names and description in section 3.2 to better reflect these differences.

From conclusion “Upon complexation with PAA, the zPCMs exhibited ~40 nm spherical micelles. These uniform assemblies were slightly larger than the PCMs formed from PEG-PLK and PEG-PVBTMA (~20-30 nm).“ However, in the text, not in the table, on the graphs, such large values for the studied micelles are not observed.

Response: We thank the reviewer for pointing out this point of confusion in reporting the size of the PCMs and zPCMs. We have made changes throughout the manuscript to clarify the Guinier radius (RGuinier) extracted from SAXS and the hydrodynamic radius (Rh) from DLS. This has been clarified by moving the multi-angle DLS data (Table 2) at 0 mM salt to show the hydrodynamic size of the micelles at no salt versus the Guinier-based size based on SAXS fitting (which likely underapproximates the size because of the difference in electron density between the core and shell of the micelles).

Why are you sure that the formed micelles are only spherical in shape?

Response: We are confident in the spheroidal shape of the micelles because of the flat (q0) scattering intensity dependency in the low-q (q ≤ 0.01) region of the SAXS profiles. In general, the power-law dependencies are observed in polyelectrolyte complex micelles (PCMs) for spheroidal (q0), rigid cylindrical (q1), and flexible cylinder (q2) form factors. We have previously demonstrated the range of form factors corresponding to imaged PCMs in previously cited publications (Lueckheide et al. Nano Lett. 2018, 18, 7111–7117; Marras, A. E. et al. Polymers 2019, 11, 83.). As an example, one of these figures is reproduced below; there is a clear q2 and q0 power law dependence in SAXS intensity for micelles that are cylinders and spheres, respectively.